# Efficacy and Safety of Biologics for Psoriasis and Psoriatic Arthritis and Their Impact on Comorbidities: A Literature Review

**DOI:** 10.3390/ijms21051690

**Published:** 2020-03-01

**Authors:** Masahiro Kamata, Yayoi Tada

**Affiliations:** Department of Dermatology, Teikyo University School of Medicine, 2-11-1 Kaga, Itabashi-ku, Tokyo 173-8605, Japan; mkamata-tky@umin.ac.jp

**Keywords:** biologics, psoriasis, psoriatic arthritis, tumor necrosis factor-α, inteleukin-23, interluekin-17

## Abstract

Psoriasis is a chronic inflammatory skin disease characterized by scaly indurated erythema. It impairs patients’ quality of life enormously. It has been recognized not only as a skin disease but as a systemic disease, since it also causes arthritis (psoriatic arthritis) and mental disorders. Furthermore, an association with cardiovascular events is indicated. With the advent of biologics, treatment of psoriasis dramatically changed due to its high efficacy and tolerable safety. A variety of biologic agents are available for the treatment of psoriasis nowadays. However, characteristics such as rapidity of onset, long-term efficacy, safety profile, and effects on comorbidities are different. Better understanding of those characteristic leads to the right choice for individual patients, resulting in higher persistence, longer drug survival, higher patient satisfaction, and minimizing the disease impact of psoriasis. In this paper, we focus on the efficacy and safety profile of biologics in psoriasis patients, including plaque psoriasis and psoriatic arthritis. In addition, we discuss the impact of biologics on comorbidities caused by psoriasis.

## 1. Introduction

Psoriasis is a chronic inflammatory skin disease characterized by scaly indurated erythema. The prevalence of psoriasis in children ranges from 0% (Taiwan) to 2.1% (Italy), and in adults it varies from 0.91% (United States) to 8.5% (Norway) [1]. It impairs patients’ quality of life enormously. It has been recognized not only as a skin disease but as a systemic disease, since it also causes arthritis (psoriatic arthritis; PsA) and mental disorders. Furthermore, an association with cardiovascular events is indicated [2]. With the advent of biologics, treatment of psoriasis dramatically changed due to its high efficacy and tolerable safety. In addition, the efficacy of biologics helps us to understand the pathogenesis of psoriasis. Whereas biologics have shown dramatically excellent efficacy, their safety has been a concern. To date, data on their efficacy and safety have been accumulated. In this paper, we focus on the efficacy and safety profile of biologics in psoriasis patients, including plaque psoriasis and PsA. In addition, we discuss the impact of biologics on comorbidities caused by psoriasis.

## 2. Efficacy and Safety of Biologics

Although approved biologic agents differ by countries, biologic agents commonly used for the treatment of psoriasis are categorized into three groups, tumor necrosis factor (TNF)-α inhibitors, interleukin (IL)-23 inhibitors, and IL-17 inhibitors, as shown in Table 1. Infliximab, adalimumab, etanercept, certolizumab-pegol, and golimumab are TNF-α inhibitors. Golimumab is used only for PsA. Ustekinumab is an anti-IL-12/23p40 antibody. Guselkumab, risankizumab, tildrakizumab, and mirikizumab are anti-IL-23p19 antibodies. Secukinumab, and ixekizumab are anti-IL-17A antibodies. Brodalumab is an anti-IL-17RA antibody. Bimekizumab is an anti-IL-17A/F antibody, which blocks both IL-17A and IL-17F. Many randomized controlled trials (RCTs) were conducted, and they demonstrated that the drugs are efficacious for moderate-to-severe plaque psoriasis. Recently, network meta-analyses enabled indirect comparison among those agents.

### 2.1. Plaque Psoriasis

Since previous review articles have illustrated the pathogenesis of psoriasis, we do not refer to the details [3,4]. Briefly, various triggers activate plasmacytoid dendritic cells, which are important in early-phase psoriasis. Meanwhile, TNF-α/iNOS-producing dendritic cells (TIP-DCs) play a pivotal role in maintaining psoriasis. TIP-DC activated by TNF-α secrete TNF-α, which activate themselves in an autocrine way. TIP-DCs also produce IL-23, which maintains and proliferates Th-17 cells. IL-23-stimulated Th17 cells are pathogenic and produce excessive IL-17A, F, and IL-22. Those cytokines drive keratinocytes to abnormal differentiation and proliferation, which forms psoriasis plaque. Activated keratinocytes also work as immune cells. They produce anti-microbial peptides and TNF-α, which activate dendritic cells including TIP-DC. This vicious inflammatory cycle makes the plaque remain and deteriorate. Activated keratinocytes secrete IL-17C, which activates keratinocytes in an autocrine way. Recently, the contribution of resident memory T cells [5,6,7,8], epidermal dendritic cells [9] in the epidermis, and type 3 innate lymphoid cells [8,10] to the development of psoriasis and recurrent plaque has attracted attention.

New biologic agents such as risankizumab [11], guselkumab [12,13,14,15,16], ixekizumab [17,18,19,20,21,22,23], and brodalumab [24,25] demonstrated high efficacy for patients with moderate-to-severe psoriasis. The percentage of patients who achieved more than a 90% reduction in the Psoriasis Area Severity Index (PASI) score at 16 weeks after initiating each treatment (PASI 90) was about 70%–80%, and that at 52 weeks was about 80%–90% in respective clinical trials. PASI 100 was about 50%–60% at 52 weeks. Those biologic agents showed high efficacy.

Ellis et al. appraised two network-meta analyses that assessed systemic therapies for moderate-to-severe psoriasis [26]. They concluded that newer biologics targeting theIL-12/23 and IL-17 axes appear to be more effective than older biologics and oral agents.

Sawyer et al. analyzed 17 studies reporting outcomes at 40-64 weeks in order to evaluate long-term efficacy [27]. Four 52-week RCTs revealed that brodalumab was significantly more efficacious than secukinumab, ustekinumab, and etanercept. Secukinumab was also more efficacious than ustekinumab, and both outperformed etanercept. Evidence from 13 additional studies and four further therapies (adalimumab, apremilast, infliximab and ixekizumab) revealed that brodalumab was most effective, followed by ixekizumab and secukinumab, then ustekinumab, infliximab and adalimumab. Etanercept had the lowest expected long-term efficacy. They concluded that brodalumab is associated with a higher likelihood of sustained PASI response, including complete clearance, at week 52 than comparators. 

Sawyer et al. performed a network meta-analysis including 77 trials (34,816 patients) [28]. They revealed the superiority of brodalumab, ixekizumab, secukinumab, guselkumab and risankizumab to tildrakizumab, ustekinumab, all TNF-α inhibitors, and non-biologic systemic treatments in efficacy for plaque psoriasis. Furthermore, brodalumab, ixekizumab, and risankizumab showed higher efficacy than secukinumab, but did not significantly compared with guselkumab. In terms of PASI 90 and PASI 100 response, brodalumab, ixekizumab, guselkumab, and risankizumab demonstrated the greatest benefits.

According to the meta-analysis of 140 studies reported by Shidian et al. [29], in terms of the percentage of patients who attained PASI 90, infliximab, ixekizumab, secukinumab, bimekizumab, brodalumab, risankizumab, and guselkumab showed higher efficacy than ustekinumab, adalimumab, certolizumab, and etanercept. Moreover, adalimumab and ustekinumab were more efficacious than certolizumab and etanercept. No significant difference was observed between any of the interventions and the placebo for the risk of severe adverse events. 

Generally, we can expect a rapid onset of efficacy on IL-17 inhibitors, especially brodalumab [30,31] and ixekizumab [32,33,34]. IL-17 inhibitors and IL-23 inhibitors, especially brodalumab, ixekizumab, and risankizumab showed high efficacy in the long-term [28,29].

### 2.2. Psoriatic Arthritis

A systematic review revealed 14.0%–22.7% of psoriasis patients having arthritis PsA in psoriasis patients [35]. PsA is characterized by enthesitis and arthritis in peripheral and axial involvement with bone proliferation and erosion [36,37]. PsA patients show distal interphalangeal joint involvement, which is usually asymmetric. In the pathogenesis of PsA, it is advocated that inflammation occurs in entheses in the early phase, and that the progression of enthesitis results in accompaniment with synovitis. In the initiation phase of enthesitis, innate immune cells are thought to play a key role. Thereafter, once certain entheseal cells are stimulated with IL-23, they secrete inflammatory cytokines such as IL-17A, IL-22 and TNF-α, consequently augmenting inflammation [36]. Focusing on enthesitis which is characteristic of PsA, Araujo et al. reported that ustekinumab, an anti-IL-12/IL-23p40 antibody, was superior to TNF-α inhibitors in the clearance of enthesitis [38], which underscores the importance of IL-23 in enthesitis. With regard to the contributions of TNF-α to the development of PsA [39,40], TNF-α increases the number of osteoclast precursor cells. TNF-α promotes differentiation into osteoclasts through RANKL-dependent pathways and promotes bone resorption. TNF-α inhibits Wnt signaling and induces apoptosis of osteoblasts, thereby reducing bone repair capacity. Recently, IL-23-independent IL-17-producing cells have drawn attention, especially in the pathogenesis of axial involvement of PsA. PsA and ankylosing spondylitis (AS) are recognized as one of spondyloarthritis (SpA) characterized by enthesitis, peripheral and axial arthritis, and negative laboratory results of rheumatoid factor and anti-cyclic citrullinated peptide antibody [41]. It indicates that the responsiveness of a certain drug for AS reflects its responsiveness for axial involvement of PsA. Anti-IL-17 antibodies, secukinumab [42] and ixekizumab [43], demonstrated efficacy for AS, but anti-IL-23 antibody, risankizumab [44], did not. Those results of clinical trials suggest the importance of IL-23-independent IL-17-producing cells in the axial involvement of PsA. iNKT and γδ-T cells are candidates of IL-23-independent IL-17-producing cells in SpA joints. Venken et al. reported that iNKT and γδ-T cells were a major source of T-cell-derived IL-17 in SpA joints and that those cells produced abundant IL-17 with anti-CD3 antibody/anti-CD28 antibody stimulation even in the absence of IL-23 [45]. Those data suggest that PsA patients with axial involvement would be better to be treated with IL-17 inhibitors such as ixekizumab and secukinumab, or TNF-α inhibitors, instead of anti-IL-23 antibodies such as risankizumab. Although the prevalence of having axial involvement in PsA patients differs by diagnosis criteria, it is reported to be from 23.6% to 55.4% [46,47]. In treating PsA patients, we should consider the considerable percentage of PsA patients having axial involvement and choose the right biologic agent. Mease et al. found the clinical characteristics of 192 PsA patients with axial involvement by comparing them with 1338 PsA patients without it [48]. Axial involvement was defined as physician-reported presence of spinal involvement at enrollment, and/or radiograph or MRI showing sacroiliitis. Patients with axial involvement showed a higher likelihood of moderate-to-severe psoriasis(body surface area ≥ 3%, 42.5% vs. 31.5%), lower prevalence of minimal disease activity (30.1% vs. 46.2%), higher nail psoriasis scores (visual analog scale; VAS 11.4 vs. 6.5), enthesitis counts (5.1 vs. 3.4), serum C-reactive protein levels (4.1 vs. 2.4 mg/L), and scores for physical function (Health Assessment Questionnaire, 0.9 vs. 0.6), pain (VAS, 47.7 vs. 36.2), and fatigue (VAS, 50.2 vs. 38.6) than patients without axial involvement. The American College of Rheumatology (ACR) and the National Psoriasis Foundation (NPF) published guidelines for PsA in 2019 [49], although nearly all recommendations were conditional since the quality of evidence was most often low or very low, and occasionally moderate.

Kawalec et al. conducted the network meta-analysis with eight eligible RCTs for efficacy (ACR 20 and ACR 50, PASI 75) and safety outcomes [50]. RCTs for abatacept, apremilast, secukinumab, or ustekinumab in adults with moderate-to-severe PsA were included. There were significant differences in the ACR20 response rate between secukinumab 150 mg and apremilast 20 mg, and between secukinumab 300 mg and apremilast 20 or 30 mg. Any adverse events occurred more often in apremilast 20 and 30 mg compared with the placebo and compared with secukinumab 150 mg. No significant differences were revealed for severe adverse events among biologics and between biologics and the placebo. In the overall population, as well as in the anti-TNF-α-naive subpopulation, secukinumab at a dose of 300 and 150 mg was ranked the highest for the ACR20 endpoint, while in the anti-TNF-α-experienced subpopulation, secukinumab 300 mg and apremilast 30 mg revealed the highest rank. For severe adverse events, the safest was ustekinumab 90 mg.

Dressler et al. also conducted a systematic review of approved systemic treatments for PsA [51]. Data were extracted from 20 trials for ACR 20, ACR 50, and adverse events after 16–24 weeks. Results for ACR 20 were infliximab + methotrexate vs. methotrexate: relative risk (RR) 1.40 (95% CI 1.07–1.84), very low-quality evidence; ixekizumab Q2W vs. adalimumab Q2W: RR 1.08 (95% CI 0.86–1.36), very low quality. In three placebo-controlled comparisons, leflunomide, methotrexate (MTX), and sulfasalazine failed to show statistical superiority for ACR. Besides the established treatment of anti-TNF antibodies and ustekinumab for psoriatic arthritis, the newer treatment options of anti-IL-17 antibodies and apremilast are also effective for the treatment of PsA. Based on just one comparative trial and one drug each, the new class of anti-IL 17 antibodies appears to be equally effective compared to the group of anti-TNF antibodies; for apremilast, this is yet unclear.

Lu et al., RCTs evaluated the efficacy and safety of targeted synthetic disease modifying anti-rheumatic drugs (DMARDs: tofacitinib, apremilast), as well as biological DMARDs (guselkumab, ustekinumab, secukinumab, ixekizumab, brodalumab, abatacept, adalimumab, etanercept, infliximab, certolizumab, and golimumab) for active PsA by a systemic literature review of 29 RCTs, including 10,204 participants and 17 treatments [52]. All treatments showed efficacy in ACR 20 and PASI 75 at 12–16 weeks in comparison with a placebo. As for safety, tofacitinib, apremilast, and ixekizumab 80 mg every 2 weeks demonstrated a higher rate of adverse events; however, in severe adverse events no significant difference was observed among all treatments. Network meta-analysis revealed that infliximab, golimumab, etanercept, adalimumab, guselkumab, and secukinumab 300 mg showed superiority to other drugs in both ACR 20 and PASI 75. Focusing on biologic-naïve patients, the results were similar to those above. Among biologic-experienced/failed patients, ustekinumab, secukinumab (300 mg and 150 mg), ixekizumab, abatacept, certolizumab pegol, tofacitinib, and apremilast still demonstrated higher ACR 20 then placebo, whereas only ustekinumab, secukinumab (300 mg), ixekizumab and tofacitinib were still associated with higher PASI 75.

In interpreting these systematic reviews, we should be aware that most clinical trials only assessed peripheral arthritis and did not assess enthesitis, dactylitis, or even axial involvement. As written above, it would be better to treat PsA with axial involvement with TNF-α inhibitors or IL-17 inhibitors instead of IL-23 inhibitors. When deciding on a treatment for PsA, we should not focus only on efficacy for arthritis, but also take skin manifestation into consideration. The safety profile should be also considered.

### 2.3. Safety Concerns of Biologics

As written above, biologic agents demonstrated tolerable safety profiles in the clinical trials. However, accumulating evidence revealed drug-specific concerns. We have already discussed the safety concerns of biologics in the previous literature [53]. Therefore, we refer to this issue in brief. It has been reported that TNF-α inhibitors are associated with serious infection (slightly increased risk), tuberculosis, paradoxical reaction, lupus, and infusion reaction (only infliximab). IL-17 inhibitors are associated with candidiasis, neutropenia, and inflammatory bowel disease. As of now, there have never been IL-23 inhibitor-specific adverse events reported. Safety concerns common in biologics include hepatitis B virus reactivation and interstitial pneumonia. Those concerns were raised during use of TNF-α inhibitors. Evidence of newer drugs such as IL-17 inhibitors and IL-23 inhibitors are insufficient. Accumulation of evidence is needed. Regarding immunogenicity, generally, predispositions of higher immunogenicity are observed in humanized monoclonal antibodies rather than in human monoclonal antibodies. The effect of immunogenicity on efficacy and safety seems limited in new biologic agents. However, since psoriasis is a chronic disease and some patients need to receive biologics for a long period of time, it might affect effectiveness and safety in the long-term. Longer observation is necessary to clarify this. With respect to malignancy, its rates are higher in psoriasis patients than the adult general population, but these treatments do not appear to increase malignancy risk.

Kaushik and Lebwohl focus on pregnant and pediatric patients with moderate-to-severe psoriasis, and those with chronic infections, such as hepatitis, HIV, and latent tuberculosis, and describe appropriate systematic treatment for them [54].

## 3. Impact of Biologics on Comorbidities

Psoriasis imposes a psychological burden on patients. The prevalence of having depression or anxiety is higher in psoriasis patients than in controls [55,56,57]. A systematic review and meta-analysis investigating 18 studies including a total of 1,767,583 participants, of whom 330,207 had psoriasis, reported that patients with psoriasis had a significantly higher likelihood of suicidal ideation, suicide attempts, and completed suicides [58]. Strober et al. evaluated the effect of biologic therapy (ustekinumab, infliximab, etanercept, and adalimumab) on depression in psoriasis patients utilizing Psoriasis Longitudinal Assessment and Registry (PSOLAR). The incidence rates of depressive symptoms were 3.01 (95% CI, 2.73–3.32), 5.85 (95% CI, 4.29–7.97), and 5.70 (95% CI, 4.58–7.10) per 100 patient-years for biologics, phototherapy, and conventional therapy, respectively. Compared with conventional therapy, biologics reduced the risk for depressive symptoms (hazard ratio, 0.76; 95% CI, 0.59–0.98), whereas phototherapy did not (hazard ratio, 1.05; 95% CI, 0.71–1.54) [59].

Cohen et al. investigated 12,502 psoriasis patients aged 20 years and above and 24,287 age- and sex-matched controls utilizing Clalit Health Services, the largest healthcare provider organization in Israel, and revealed that psoriasis was associated with Crohn′s disease (CD; odds ratio, 2.49; 95% CI, 1.71–3.62) as well as ulcerative colitis (UC; odds ratio, 1.64; 95% CI, 1.15–2.33) [60]. Focusing on the effect of biologics, randomized control trials for CD with secukinumab and brodalumab resulted in a disproportionate number of cases of worsening and no evidence of efficacy [61,62]. In addition, exacerbations of inflammatory disease have been reported in psoriasis patients receiving IL-17 inhibitors [22,25,60]. Lee et al. found that IL-17A-dependent regulation of the tight junction protein occludin in the intestinal mucosa during epithelial injury limits excessive permeability and maintains barrier integrity, and revealed that IL-23-independent IL-17 production by γδ T cells was important for the maintenance and protection of epithelial barriers in the intestinal mucosa. Whereas IL-17 inhibition exacerbates inflammatory bowel disease, TNF-α inhibitors are approved for the treatment of CD and UC, and IL-23 inhibitors demonstrated clinical improvement [63,64].

Recently, “psoriatic march”, the concept of a causal link between psoriasis and cardiovascular disease, has been widely recognized. Systemic inflammation may cause insulin resistance, which in turn triggers endothelial cell dysfunction, leading to atherosclerosis and finally myocardial infarction or stroke [65,66]. Systematic review and meta-analysis analyzing 14 papers, including a total of 25,042 patients with psoriasis, revealed the association of psoriasis with metabolic syndrome. They reported that metabolic syndrome was present in 31.4% of patients with psoriasis (odds ratio, 1.42; 95% CI, 1.28–1.65) [67]. Other articles indicated that obesity is associated with the onset, exacerbation, and intractability of psoriasis [68,69,70,71,72]. As for hyperglycemia, Ikumi et al. revealed that hyperglycemia is highly associated with psoriasis, mainly through IL-17. In patients, the severity of psoriasis correlated with high blood glucose levels, and anti-IL-17A monoclonal antibody therapy reduced HbA1c levels significantly in these patients. In imiquimod-induced psoriasiform dermatitis, treatment with anti-IL-17A monoclonal antibody decreased fasting blood glucose levels [73]. Concerning endothelial cell dysfunction, flow-mediated dilation (FMD) is a marker that reflects endothelial cell dysfunction. Avgerinou et al. investigated FMD in 14 psoriasis patients before and 12 weeks after treatment with adalimumab, and reported that it improved after treatment with adalimumab [74]. von Stebut et al. conducted a 52-week, randomized, double-blind, placebo-controlled, exploratory trial in patients with moderate-to-severe plaque psoriasis without clinical cardiovascular disease, named Evaluation of Cardiovascular Risk Markers in Psoriasis Patients Treated with Secukinumab (CARIMA) [75]. Although a statistical difference was not observed in the baseline-adjusted mean FMD between patients receiving secukinumab and those receiving placebo at week 12, FMD was significantly higher than baseline in patients receiving the label dose of 300 mg secukinumab for 52 weeks. Regarding atherosclerosis, Hjuler et al. examined calcified coronary plaque, utilizing cardiac computed tomography angiography in severe psoriasis patients, severe atopic dermatitis patients, and retrospectively matched controls [76]. They demonstrated that psoriasis patients showed an increased prevalence of severe coronary stenosis (stenosis >70%) (psoriasis 14.6%, controls 0%; *p* = 0.02) and 3-vessel coronary affection or left main artery disease (psoriasis 20%, controls 3%; *p* = 0.02), whereas AD patients showed an increased prevalence of mild single-vessel affection (AD 40.7%, controls 9.1%; *p* = 0.005). Elnabawi et al. conducted a prospective, observational study in order to investigate the effect of biologic therapy on coronary artery plaque [77]. Analysis of 121 participants who were biologics-naïve at baseline and received biologic therapy for one year revealed that biologic therapy was associated with a 6% reduction in non-calcified plaque burden (*p* = 0.005) reduction in necrotic core (*p* = 0.03), with no effect on fibrous burden (*p* = 0.71), indicating that biologic therapy in severe psoriasis was associated with favorable modulation of coronary plaque indices. After one-year of biologic therapy, non-calcified plaque burden decreased by 5% in patients treated with TNF-α inhibitors (*p*  = 0.06), by 2% in patients treated with anti-IL12/23 antibody (*p* = 0.36), and by 12% in patients treated with IL-17 inhibitors (*p* < 0.001). Patients treated with IL-17 inhibitors demonstrated a significantly greater reduction in non-calcified coronary plaque burden compared with those treated with anti-IL12/23 antibody and those with no biologic treatment. Patients treated with TNF-α inhibitors showed a significantly greater reduction in non-calcified coronary plaque burden only compared those with non-biologic treatment (*p* < 0.01). Regarding cardiovascular events, a systemic review and meta-analysis demonstrated that mild and severe psoriasis are associated with an increased risk of myocardial infarction and stroke, and that severe psoriasis is also associated with an increased risk of cardiovascular mortality [78]. Yang et al. conducted a meta-analysis on the effect of TNF-α inhibitors on cardiovascular events in psoriasis and psoriatic arthritis, analyzing five studies (49,795 patients). Compared with topical/photo treatment, TNF-α inhibitors were associated with a significant lower risk of cardiovascular events (RR, 0.58; 95% confidence interval; CI, 0.43 to 0.77; *p* < 0.001). Additionally, compared with MTX treatment, risk of cardiovascular events was also markedly decreased in the TNF-α inhibitor group (RR, 0.67; 95% CI, 0.52 to 0.88; *p* = 0.003). Wu et al. also examined patients receiving TNF-α inhibitors (*n* = 9148) and patients receiving MTX (*n* = 8581). Psoriasis patients receiving TNF-α inhibitors had a lower major cardiovascular event risk compared to those receiving MTX (Kaplan–Meier rates: 1.45% vs. 4.09%: *p* < 0.01. Hazard ratio = 0.55; *p* < 0.01) [79]. The direct effect of IL-17 inhibitors or IL-23 inhibitors on cardiovascular events has not been reported yet.

Kaushik and Lebwohl also described specific comorbidities and insights to choose the appropriate systemic treatment in patients with moderate-to-severe psoriasis [80]. The choice of appropriate biologic therapy for a patient is often determined by the presence of comorbidities.

## 4. Conclusions

Evidence of new drugs on long-term efficacy, safety, and impacts on comorbidities is relatively low. Further accumulation is needed to clarify them. Currently, a variety of biologic agents are available for the treatment of psoriasis. However, characteristics such as rapidity of onset, long-term efficacy, safety profile, and effects on comorbidities are different. Better understanding those characteristics leads to the right choice for individual patients, resulting in higher persistence, longer drug survival, higher patient satisfaction, and minimizing the disease impact of psoriasis.

## Figures and Tables

**Table 1 ijms-21-01690-t001:** Target of biologics.

Target of biologics	Drug
TNF-α inhibitors	Infliximab
	Adalimumab
	Golimumab (for psoriatic arthritis)
	Certolizumab-pegol
	Etanercept
IL-12/23 inhibitor	Ustekinumab
IL-23 inhibitors	Guselkumab
	Risankizumab
	Tildrakizumab (not approved yet)
	Mirikizumab (not approved yet)
IL-17 inhibitors	Secukinumab
	Ixekizumab
	Brodalumab
	Bimekizumab (not approved yet)
CTLA4-Ig	Abatacept (for psoriatic arthritis)

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
