# Peer review of "Efficacy and Safety of Biologics for Psoriasis and Psoriatic Arthritis and Their Impact on Comorbidities: A Literature Review"

_ijms, 2020, doi:10.3390/ijms21051690_

Round 1

Reviewer 1 Report

This is a well-written article. I have one suggestion:

Section 3. Impact of biologics on comorbidities - at the end of this section, please also mention that very often the choice of appropriate biologic therapy for a patient is determined by the presence of comorbidities. For instance, Infliximab and Ustekinumab are the preferred biologics for psoriasis with obesity owing to their weight-based dosing regimen. 

Please refer to the following articles for additional details:

www.ncbi.nlm.nih.gov/pubmed/30017706

www.ncbi.nlm.nih.gov/pubmed/30017705 

Author Response

This is a well-written article. I have one suggestion:

Section 3. Impact of biologics on comorbidities - at the end of this section, please also mention that very often the choice of appropriate biologic therapy for a patient is determined by the presence of comorbidities. For instance, Infliximab and Ustekinumab are the preferred biologics for psoriasis with obesity owing to their weight-based dosing regimen. 

Response: We appreciate the comments. Following the reviewer’s suggestion, we added the sentences as below.

“Kaushik S and Lebwohl M also described specific comorbidities and insights to choose appropriate systemic treatment in patients with moderate-to-severe psoriasis (J Am Acad Dermatol. 2019 Jan;80(1):27-40). The choice of appropriate biologic therapy for a patient is often determined by the presence of comorbidities.”

As for weight-based dosing regimen, recent articles revealed that ixekizumab, brodalumab, and risankizumab showed efficacy regardless of body weight or BMI. And those biologic agents demonstrated greater efficacy than infliximab and ustekinumab 90mg. Therefore, we did not mention this issue.

Please refer to the following articles for additional details:

www.ncbi.nlm.nih.gov/pubmed/30017706

Response: We added the sentences and reference at the end of “Safety concerns of biologics” section as follows: “Kaushik S and Lebwohl M focus on pregnant and pediatric patients with moderate-to-severe psoriasis, and those with chronic infections, such as hepatitis, HIV, and latent tuberculosis and describe appropriate systematic treatment for them (J Am Acad Dermatol. 2019 Jan;80(1):43-53.).”

www.ncbi.nlm.nih.gov/pubmed/30017705 

Response: We added the sentences and reference at the end of “Impact of biologics on comorbidities” section as follows: “Kaushik S and Lebwohl M also described specific comorbidities and insights to choose appropriate systemic treatment in patients with moderate-to-severe psoriasis (J Am Acad Dermatol. 2019 Jan;80(1):27-40). The choice of appropriate biologic therapy for a patient is often determined by the presence of comorbidities.”

Reviewer 2 Report

The paper presents a summary of the most important studies and meta-analyzes regarding the efficacy and safety of biological treatment of psoriasis and psoriatic arthritis. 

From a methodological point of view, the work is correct and contains current
knowledge. In recent years, however, a lot of reviews and meta-analyses are
being done on the biological treatment of psoriasis and psoriatic arthritis.
Unfortunately, but in this field the work adds nothing new and has limited
novelty.However, since it was wualified for review, I belive that the topic
of the manuscript was of interest to the journal.It can be accepted in current
form.

Author Response

We appreciate the reviewer’s comments.